# Optimization of 4,6-α and 4,3-α-Glucanotransferase Production in *Lactococcus lactis* and Determination of Their Effects on Some Quality Characteristics of Bakery Products

**DOI:** 10.3390/foods13030432

**Published:** 2024-01-29

**Authors:** Ramazan Tolga Niçin, Duygu Zehir-Şentürk, Busenur Özkan, Yekta Göksungur, Ömer Şimşek

**Affiliations:** 1Department of Food Engineering, Faculty of Chemical and Metallurgical Engineering, Yıldız Technical University, Istanbul 34220, Turkey; nicin.ramazan@gmail.com; 2Department of Food Engineering, Faculty of Engineering, Pamukkale University, Denizli 20160, Turkey; dzehir@pau.edu.tr (D.Z.-Ş.); ozkanbusenur@hotmail.com (B.Ö.); 3Department of Food Engineering, Faculty of Engineering, Ege University, İzmir 35100, Turkey; yekta.goksungur@ege.edu.tr

**Keywords:** *Lactococcus lactis*, α-glucanotransferase, bread, bakery products, glycemic index, staling

## Abstract

In this study, the production of 4,6-α (4,6-α-GTase) and 4,3-α-glucanotransferase (4,3-α-GTase), expressed previously in *Lactococcus lactis*, was optimized and these enzymes were used to investigate glycemic index reduction and staling delay in bakery products. HP–SEC analysis showed that the relevant enzymes were able to produce oligosaccharides from potato starch or malto-oligosaccharides. Response Surface Methodology (RSM) was used to optimize enzyme synthesis and the highest enzyme activities of 15.63 ± 1.65 and 19.01 ± 1.75 U/mL were obtained at 1% glucose, pH 6, and 30 °C for 4,6-α-GTase and 4,3-α-GTase enzymes, respectively. SEM analysis showed that both enzymes reduced the size of the starch granules. These enzymes were purified by ultrafiltration and used to produce bread and bun at an enzyme activity of 4 U/g, resulting in a decrease in the specific volume of the bread. It was found that the estimated glycemic index (eGI) of bread formulated with 4,6-α-GTase decreased by 18.01%, and the eGI of bread prepared with 4,3-α-GTase decreased by 13.61%, indicating a potential delay in staling. No significant differences were observed in the sensory properties of the bakery products. This is the first study showing that 4,6-α-GTase and 4,3-α-GTase enzymes have potential in increasing health benefits and improving technological aspects regarding bakery products.

## 1. Introduction

Recently, some lactic acid bacteria (LAB) have been found to possess α-glucanotransferase (α-GTase) enzymes with catalytic effects on maltodextrin and starch [1,2,3,4]. Two different α-glucanotransferases (4,6-α-GTase and 4,3-α-GTase) synthesized by LABs show specific hydrolase/trans-glycosylase activity toward malto-oligosaccharides (MOS) and starch [5,6,7,8].

These enzymes contain eight (β/α) barrel structures, four conserved amino acid sequence motifs, Glu and Asp amino acids in their active sites, and show catalytic activity via an α-retaining mechanism. AGTases act on substrates containing several consecutive (α1→4) glycosidic bonds, such as amylose, amylopectin, maltodextrins and glycogen. A critical feature of AGTases is that they catalyze the disproportionation reaction and transfer the cleaved glucan to a glucan acceptor that forms a new α-glycosidic bond [9,10,11,12].

This catalytic reaction begins with the cleavage of a (α1→4) glycosidic bond in the substrate and the formation of an α-bonded covalent glycosyl-enzyme intermediate. Following detachment of the glucan fragment from the acceptor subunit, the non-reducing end of another glucan can enter the acceptor subunit and attack the hydroxyl group located at different carbon atoms (usually 4, 6, or 3) to form new (α1→4), (α1→6), or (α1→3) glycosidic bonds. The genetic segment corresponding to Gtase I exhibited catalytic activity in the presence of maltodextrin and starch. This enzyme was identified as 4,6-α-GTase because it showed the capacity to create (α1→6) branches by cleaving (α1→4) bonds. In the following studies, it was determined that L. reuteri NCC 2613 produced a similar 4,6-α-GTase and *L. fermentum* NCC 2970 produced a 4,3-α-GTase, which hydrolyze (α1→4) bonds and form (α1→3) branches. Following their biochemical characterization, α-glucanotransferases in LABs were designated as GtfB [1,5,6,13].

The GtfB enzyme of *L. reuteri* 121 strain formed new products with 91 and 61% (α1→6) branching at (α1→6) ratios when amylose V (average Mw = 200 kDa) and high amylose starches (amylose content > 52%) from potato starch were used as substrates [3,14].

Studies on the role of amylose and amylopectin molecules in retrogradation suggest that retrogradation consists of two separate processes: rearrangement of amylose, which becomes soluble during gelatinization, and recrystallization of amylopectin from gelatinized granules [15]. In general, long-term changes characterized by an increase in crystallinity are associated with amylopectin, whereas short-term changes in the initial stages are associated with amylose [16]. Within the framework of the problem described above, a strong approach is to delay starch retrogradation using 4,6-α-GTase and 4,3-α-GTase enzymes to break the (α1→4) glycosidic bonds in the amylose units in the starch molecule, causing a change in the ratio of amylose and amylopectin by forming (α1→6) bonds.

Li et al. (2018) observed that, when wheat starch was treated with 4 U/g 4,6-α-GTase enzyme at pH 6 for 40 °C/1 h, the molecular size decreased from 107 to 106 kDa, the amylose content decreased from 22.07% to 17.11% and the amylopectin long branches decreased from 26.4% to 15.6%. In addition, when the enzyme-modified starches were gelatinized and kept at 4 °C for 1–2 weeks after gelatinization, the endothermic enthalpy values were found to be lower than those of the unmodified starches, and it was concluded that the 4,6-α-GTase enzyme inhibited starch retransformation [17].

Although the effect of 4,6-α-GTase and 4,3-α-GTase produced by LABs on bread properties are unknown, enzymes belonging to closely related families (α-glucanotransferase, maltogenic amylase, pullulanase, etc.) have been found to be effective in delaying the staling of bakery products [18,19].

In our previous study, 4,6-α-GTase and 4,3-α-GTase enzymes from the two lactic acid bacteria, *Limosilactobacillus reuteri* E81 and *Limosilactobacillus fermentum* PFC282, were successfully cloned and expressed in *L. lactis* [20]. The aim of this research was to produce previously expressed extracellular heterologous glucanotransferases, with expected application in retarding staling and reducing the glycemic index of bakery products.

## 2. Materials and Methods

### 2.1. Recombinant Lactococcus lactis and Enzyme Production

4,6-α-GTase and 4,3-α-GTase genes from *Limosilactobacillus reuteri* E81 (Accession number: OP846570) and *Limosilactobacillus fermentum* PFC282 (Accession number: OP846569), respectively, were transformed into the plasmid *pLEB*124 vector containing the signal peptide *usp*45 under the *p*45 continuous promoter and successfully expressed in *Lactococcus lactis* MG1363, as described previously in our study [20]. Enzymes were produced in M17 medium (Merck, Darmstadt, Germany) containing 0.5% glucose (Merck, Germany) by propagating recombinant *L. lactis* for 48 h at 30 °C.

### 2.2. Enzyme Activity Assay

The activity of purified glucanotransferases was determined using different substrates. For the amylose-iodine test, a stock solution (260-fold) was prepared by dissolving 0.26 g of iodine and 2.6 g of potassium iodide (KI) in 10 mL of water. After dilution, 150 μL of this solution was mixed with the reaction medium and incubated with 15 μL enzyme. The optical density of the reaction medium was recorded at OD_660_. To conduct another test with amylose, a 1% Amylose V (AVEBE, Veendam, The Netherlands) stock solution was prepared by dissolving 20 mg Amylose V in 1 mL distilled water and 1 mL of 2 M NaOH. An equivalent volume of 1 M HCl was added to the reaction medium to ensure neutralization immediately before the activity test. The reaction mixture for activity testing, utilizing a stock solution, consisted of 0.25% Amylose V, 25 mM NaAc, and 1 mM CaCl_2_. The enzyme, at a concentration of 2.5 μg mL^−1^, was introduced into the reaction medium, followed by incubation at 37 °C. The control of the resulting products was recorded using an amylose–iodine test [21].

### 2.3. Production and Optimization of 4,6-α-GTase and 4,3-α-GTase under Bioreactor Conditions

The cultivation of *Lactococcus* lactis cells transfected with the *gtfB* gene for the production of 4,6-α-GTase and 4,3-α-GTase was carried out via batch fermentation in a 2-L bioreactor (Infors HT, Basel, Switzerland) equipped with pH (Mettler-Toledo LLC, Colombus, OH, USA), temperature, and aeration (Mettler Toledo, USA) control systems. The optimal initial pH and temperature values for the bioreactor were determined using a shaker flask system. Both cell density and biomass quantities were monitored during the fermentation period in bioreactor experiments. Following cell separation by centrifugation, enzyme activity in the cell supernatant was determined using the amylose–iodine method.

### 2.4. Purification of 4,6-α-GTase and 4,3-α-GTase

To purify 4,6-α-GTase and 4,3-α-GTase produced extracellularly at the end of batch fermentation under optimum conditions, biomass separation was performed by centrifugation (Himac CR22N, Hitachi, Japan) at 6000× *g* for 15 min. During enzyme purification, enzyme activity was determined according to the amylose–iodine method in triplicate [21].

Sartocon Slice 200 (Sartorius Stedim Biotech, Goettingen, Germany) was used for enzyme purification by ultrafiltration (UF). For this purpose, the fermentation liquid was passed through the separation boundaries of 300 and 100 kDa. After the first 300 kDa ultrafiltration, the permeate was applied to the other filter. Before the UF process, 2 L of pure water was passed through the system at the operating pressure values to wet the membrane surface, 2 L initial volume of enzyme-containing preparation was used, and ultrafiltration continued until 1.8 L permeate and 0.2 L retentate was collected. Thus, UF was realized with 90% permeate and 10% retentate ratio by volume. Because the molecular size of the enzymes was larger than 100 kDa, the retentate obtained after the 100 kDa membrane was used for liquid enzyme preparation. The enzyme activity in the retentate was determined using the amylose–iodine method.

### 2.5. Lyophilization of 4,6- and 4,3-α-Glucanotransferase Enzymes

In the two strategies used to prepare the enzymes, different concentrations of ethanol, acetone, or ammonium sulfate were used to precipitate the enzyme in the fermentation supernatant. For this purpose, experiments were carried out at four different concentrations, ranging from 10–50% for ethanol and acetone and 20–50% for ammonium sulfate precipitation.

Freeze-drying was used to concentrate and preserve the enzyme preparations. For this purpose, different cryogenic substances were added to the enzyme solutions, and the stability of the enzyme was investigated. Four different formulations were tested: 200 mM sucrose, 200 mM sucrose + 1% BSA (Bovine Serum Albumin), 200 mM sucrose + 0.5% gelatin, and 200 mM sucrose + 1% PEG (polyethylene glycol), respectively. The prepared lyophilization stabilizers were mixed with the enzyme preparations in equal proportions and then freeze-dried. A laboratory freeze dryer (Thermo Fischer, Waltham, MA, USA) was used, and the working conditions were −50 °C and 200 mTorr for 8 h. The product obtained was stored in a modified atmosphere by filling special bottles, and enzyme activity analysis was performed according to the amylose–iodine method to determine the efficiency of freeze-drying.

### 2.6. SEM Images of the Effect of 4,6-α-GTase and 4,3-α-GTase on Starch

The morphologies of the products formed by the catalytic effects of 4,6-α-GTase and 4,3-α-GTase enzymes on starches from different sources (wheat, corn and potato) were determined. The morphology of the starch granules was determined using scanning electron microscopy (SEM, S-4800, Hitachi, Japan). The microscopic morphology and structure of starch granules were observed. Sample imaging was performed at an accelerating voltage of 20.00 kV and amplification factors of 5000, 20,000, or 60,000.

### 2.7. Utilization of 4,6-α-GTase and 4,3-α-GTase in Bakery Products and Determination of Their Effects on Quality Characteristics

The production of two different 4,6-α-GTase and 4,3-α-GTase enzymes produced in the study was effective in delaying staling and reducing the glycemic index in bakery products, bread and bun, which are widely consumed at a significant rate within the category. Two experimental samples were used for this purpose. The first sample contained lyophilized 6-α-GTase and 4,3-α-GTase (4 U/g flour) [17], whereas the second sample was a control sample without enzymes. Each production step was repeated three times.

### 2.8. Bread Production Using GtfB Enzymes

The amount of water used in the production was 2% less, considering the water-lifting capacity determined by farinograph analysis. To avoid disturbing the liquid phase equilibrium, the amount of liquid enzyme (mL) added according to the experimental design was deducted from the amount of water. The flour-based ingredients listed in Appendix A were kneaded in a mixer (KitchenAid, Benton Harbor, MI, USA) for 10 min, using the direct dough method. The resulting dough was cut into pieces of 50 ± 2 g. The doughs were fermented in a fermentation cabinet at a relative humidity above 80% and 30 °C for 30 min. Following manual removal of gas and roll shaping, the dough was transferred to pans and allowed to undergo final fermentation. The duration of the final fermentation was determined when the dough rose by 1.5 cm above its initial height, equivalent to 90% of its natural rise. The samples were baked in an air circulation oven (ASL, APF-50 Model, Konya, Turkey) at 200 °C for 15 min.

### 2.9. Bun Preparation Using GtfB Enzymes

The ingredients listed in Appendix A were used for the production of the buns. Flour, shortening (anhydrous vegetable shortening), sugar, and salt were mixed in a mixer until the mixture became lumpy. Eggs, yeast, and water were added and kneaded for 4 min to obtain an unhomogenized dough sample. Dough was cut into 50 ± 2 g pieces and rounded after 15 min of fermentation (85% at 30 °C). After 5 min of fermentation, they were shaped like buns and placed on trays. After 90 min of final fermentation, the egg yolk was applied to the surface with a brush and the mixture was baked in an oven at 200 °C for 20 min.

### 2.10. Textural Properties of Bakery Products

Texture profile analysis of the bread rolls was performed using a texture analyzer (Model Nu: CT3-4500, Brookfield Amatek, Essex, UK) with a cylinder probe. Slices of 30 mm were cut from the bread, and measurements were made at a speed of 5 mm/s, plunge depth of 10 mm (approximately 33% deformation), and initial sensing force of 5 g. The hardness, adhesive and cohesive adhesiveness, flexibility, elasticity, and chewability properties of the bread rolls were also examined [22].

Hardness (g), adhesive and cohesive adhesiveness, flexibility, elasticity, and chewability were determined according to Standard Method 74-09 [23]. A horizontal section of 1.25 cm thickness was taken and measured from the center. Measurements were performed using a cylinder probe with a speed of 1.7 mm/sec, plunge depth of 10 mm, and initial sensing force of 5 g [24]. In addition, bread and bun samples were lyophilized, and the surface differences between bread and bun were determined and compared to those of the control samples.

### 2.11. Determination of Estimated-Glycemic Index (eGI) Values

The estimated glycemic index (eGI) of the bread samples was determined according to the methods described previously [25,26]. To determine the glycemic index in vitro, 1 g of homogenized bread was mixed with 5 mL of deionized water. Pepsin–guar gum solution (10 mL) was added to the sample and incubated at 37 °C for 30 min in a shaking water bath (175 rpm). After incubation, 0.5 M sodium acetate solution (5.0 mL) was added and the pH was adjusted to pH 5.0–5.25. An enzyme solution containing pancreatin and amyloglycosidase (Novozymes, Bagsvaerd, Denmark) (13.4 U/mL) was added, and the volume was adjusted to 50 mL with deionized water. The samples were then incubated in a shaking water bath for 180 min. During incubation, samples (0.5 mL) were collected at 20, 30, 60, 90, 120, and 180 min and placed in separate test tubes. Test tubes were placed in a boiling water bath for 5 min to ensure enzyme denaturation. The final volume was adjusted to 5 mL using deionized water and the samples were centrifuged at 4000× *g* for 5 min. The glucose content of the supernatant was determined using a GOPOD-formatted K–GLUC assay kit (Megazyme, Wicklow, Ireland) with a spectrophotometer (Shimadzu UV-1800, Kyoto, Japan) at a wavelength of 510 nm.

The eGI was calculated from the hydrolysis index (HI) value of each sample. The HI value was obtained by dividing the area of white bread obtained from the local market by the area under the hydrolysis curve, eGI was calculated described by [27].

### 2.12. DSC Analysis of Bakery Products

In accordance with the experimental design, enthalpy values were measured by differential scanning calorimetry (DSC) to determine the retarding properties of the enzymes on the staling properties of the samples. On days 0, 3, and 7, roll bread and bun samples stored at 4 °C were analyzed using DSC (Perkin Elmer Pyris 6 DSC). The device was calibrated with indium (156.6 °C, 28.591 J/g) and tin (232.2 °C, 60.62 J/g), while distilled water was used as a reference. Samples (10 mg) were weighed hermetically and accurately in aluminum containers. The container was exposed to heat from 20 °C to 130 °C with an increase of 5 °C/min, and the enthalpy values were calculated by calculating the areas between 40 °C and 80 °C [28].

### 2.13. Sensory Analysis

A panel of trained individuals conducted the sensory analysis. The panelists were chosen following a preliminary selection process based on their ability to distinguish six basic tastes: sweet (sucrose), salty (NaCl), sour (citric acid), bitter (crystallized caffeine), metallic (iron II sulfate heptahydrate), and umami (monosodium glutamate). The selected 10 panelists were trained to improve their ability to evaluate bread and bun products in terms of sensory properties. In the sensory evaluation trials, breads were sliced (thickness 15 mm), and buns were divided into four pairs, coded randomly with three numbers, and presented to the panelists.

Rolled bread samples were evaluated in terms of color, pore structure, internal texture, chewiness, softness, wetness–dryness, swallowing, fermented odor, acid odor, and general evaluation using a linear unipolar scale consisting of 1 (very bad) to 10 (very good) boxes [29].

### 2.14. Statistical Analysis

#### 2.14.1. Experimental Design and Optimization

The effects of different process parameters (initial substrate concentration, aeration rate (vvm), and pH) on synthesis of 4,6-α-GTase and 4,3-α-GTase enzymes were investigated using the response surface methodology (RSM). The parameter levels were determined through preliminary experiments. Box–Behnken design (BBD) was used to improve recombinant GtfB production, considering the optimized levels of variables [30]. The initial substrate concentration ($X_{1}$, (g dm^−3^, %), Air-flow rate ($X_{2}$, vvm) and pH ($X_{3}$) were chosen as independent variables (Table 1). In this study, a face-centered statistical design (face central statistical design, α = 1), with 15 experimental points and three variables, was used. The levels are −1, 0, and +1, with 0 being the center point. Fifteen experiments were conducted in triplicate and the average was used as the response. Regression analysis was performed on the experimental results of the BBD, and the data were used to fit a quadratic equation. The response was analyzed using the second-order polynomial in Equation (1):(1)$$Y=\beta_{0}\overset{k}{\underset{i=1}{\sum }}\beta_{i}X_{i}+\overset{k}{\underset{i=1}{\sum }}\beta_{ii}X_{i}^{2}+\overset{k}{\underset{i<1}{\sum }}\beta_{ij}X_{i}X_{j}$$

After the levels of the parameters were determined, the linear, quadratic, and interaction effects of the parameters on enzyme activity were determined using Response Surface Methodology, and the optimum levels of the parameters providing the highest enzyme activity were determined using the model. The Desing Expert 12 statistical package program was used to create the model and the test plan provided by the program was implemented. A mathematical model was created using the experimental data, and the optimal conditions for the process were determined.

The average maximum recombinant 4,6-α-GTase and 4,3-α-GTase activities were used as the dependent variable or response (*Y*). A second-order polynomial equation was then fitted to the data using multiple regression analysis, and an empirical model that related the response evaluated for the independent variables of the experiment was obtained.

The quadratic model obtained from the regression analysis allowed the construction of a three-dimensional (3D) graph in which the dependent variable *Y* was represented by a curvature surface as a function of $X_{i}$. The relationship between the response and independent variables can be directly seen from the response surface plot. The statistical software Design Expert v.12 (Minneapolis, MN, USA) was used to establish the experimental design and regression analysis of the experimental data (Table 2). The fit of the polynomial model equation is expressed by the coefficient of determination R^2^ [31].

#### 2.14.2. Statistical Analysis of Data

Statistical differences (*p* ≤ 0.05) between the samples in terms of textural properties, glycemic index and enthalpy values were analyzed by one-way analysis of variance (ANOVA) using the statistical program Design Expert 12. Differences between groups were determined using Tukey’s multiple comparison method. The illustration of the RATA feature relates to a rate-all-that-apply (RATA) experiment conducted using the XLSTAT v. 2023 5.2 (Denver, CO, USA).

## 3. Results and Discussion

### 3.1. Structural Characterization of Products Formed by the Catalytic Action of 4,6-α-GTase and 4,3-α-GTase Enzymes

Previously, we expressed two glucanotransferases extracellularly in *L. lactis* and characterized the properties of the purified glucanotransferases [20]. All plasmids constructed in our previous study, pLEB124: gtfB1 (2666 bp) and pLEB124: gtfB2 (3243 bp), contained the usp45 signal sequence to enable *Lactococcus* spp. to secrete heterologous extracellular proteins. We attempted to express this gene in Lactococcus lactis cells without signal transduction. The pLEB124 vector was constructed containing the 2666 bp part of the gene, and this plasmid contained a continuous promoter p45 to increase the efficiency of extracellular production. The resulting recombinant plasmid was transferred to the *L. lactis* MG1363 host and *L. lactis* pLAC46, capable of producing the 4,6-α-GTase enzyme extracellularly. The 3243 bp of the 4,3-α-GTase enzyme was successfully cloned into the pLEB124 vector by adding the usp45 signal peptide to its N-terminal region. The resulting recombinant plasmid was transferred to the *L. lactis* MG1363 host, and *L. lactis* pLAC43 cells capable of extracellular production of 4,3-α-GTase enzyme were obtained. After colony PCR, the genes of interest were verified by sequencing and further biochemical studies were performed. After the expression of glucanotransferases and their purification, SDS-PAGE, western blotting, and NMR analysis were performed to determine the protein biochemistry in our previous study [20].

As seen in high-pressure size exclusion chromatography (HPSEC) chromatograms, the samples containing 0.0025% potato starch, 1% 4,3-α-GTase and 1% 4,6-α-GTase had peak values of 0.554 uRIU, 0.189 uRIU and 0.173 uRIU, respectively at 12.5–15 min (Figure 1).

Thus, the decrease in the enzyme-reacted starch values compared to the control confirmed the (α1→4) bond hydrolase/transglycosylase activity of the 4,6-α-GTase and 4,3-α-GTase enzymes in starch. After this activity, the peak values between 15 and 17.5 min were determined to correspond to new oligosaccharide forms. It was found that 4,6-α-GTase and 4,3-α-GTase enzymes acted on the α(1→4) bonds in starch. Thus, it was concluded that maltodextrin structures smaller than the amylose structure were formed, as reported previously [7,14].

### 3.2. Extracellular Production and Optimization of GtfB Enzymes in Bioreactor

*Lactococcus lactis* is widely recognized in the field of biotechnology as a highly efficient “bioreactor” for protein production. Hence, various vectors containing constitutive or inducible promoters have been developed for heterologous protein expression in *Lactococcus lactis*. Among these, p45 and p32 promoters are particularly noteworthy and can be used for different purposes [32]. Additionally, the GRAS status of *L. lactis* strains and expression of novel enzymes such as 4,6-α-GTase and 4,3-α-GTase are important factors to consider when using these strains as hosts. To develop a bioreactor environment for the extracellular production of these enzymes through this host, the most suitable initial pH and temperature values for the bioreactor system were found to be 6 and 30 °C, respectively, in a shaking flask system. In studies carried out in bioreactors, the amount of biomass was measured during the fermentation period, and enzyme activity was determined using the amylose–iodine method in the cell supernatant after cell separation by centrifugation.

#### 3.2.1. The Results of BBD Experiments of 4,6-α-GTase and 4,3-α-Gtase

A Box Behnken Design (BBD) experiment with three factors (N = 15) was carried out to increase the synthesis of 4,6-α-GTase and 4,3-α-GTase enzymes. The central points of initial substrate concentration, 3%; pH 5.5; and air-flow rate, % 30 (0.1 vvm) were chosen. The experimental results are listed in Table 2.

The highest 4,6-α-GTase (15.63 ± 1.65 U/mL) and 4,3-α-GTase yields (19.01 ± 1.75 U/mL) were obtained in run 1. In another study, the specific total activity of the purified *L. fermentum* NCC 2970 4,3-α-GTase enzyme on 0.125% (wv^−1^) amylose V was determined as 22 ± 0.36 U mg^−1^ protein by using an iodine-staining assay [7]. Bai et al. (2015) determined the total activities of GtfB-N as 2.8 U/mg using the optimal conditions pH 5.0 and 40 °C [21]. In another study on the enzyme GtfZ, which makes (α1→3)-branching activity on IMMP, GtfZ-CD2 had clear transglucosylating activity on both dextran and IMMP. A lower initial rate was observed with IMMP, 10.4 and 14.8 U/mg protein for IMMP (Mw 18.3 kDa) and dextran (Mw 70 kDa), respectively [33]. In our study, the highest 4,6-α-GTase yield was observed with 15.63 ± 1.65 U/mL, while the highest 4,3-α-GTase yield was observed with 19.01 ± 1.75 U/mL.

In the production of the 4,6-α-GTase enzyme in a fermenter, it was found that pH and the initial substrate concentration were the two factors that had the greatest impact on enzyme yield (*p* < 0.05). In 2011, Kralj et al. showed that the incubation of GtfB with a large amylose-type donor substrate (amylose-V) and smaller saccharides (glucose and maltose) as acceptor substrates resulted in the synthesis of larger saccharides with both (α1→6) and (α1→4) glycosidic linkages [1]. Thus, in our study, glucose, which was used as the initial substrate, had a statistically significant impact on the yield of 4,6-α-GTase, whereas it failed to exhibit a statistically discernible influence on the production of 4,3-α-GTase [1]. However, pH was the sole factor affecting the yield of 4,3-α-GTase (*p* < 0.05).

In the BBD experiment, the quadratic regression equation affecting the production of 4,6-α-GTase and 4,3-α-GTase was determined using Design Expert 12.0, as in the equation in 4,6-α-GTase (Equation (2)):

4,6-α-GTase Yield:(2)$$\begin{array}{c} =8.523-1.235\;X_{1}-0.354\;X_{2}+2.711\;X_{3}+1.262\;X_{1}^{2}+1.630\;X_{2}^{2}-0.175\;X_{3}^{2}\\ -0.145\;X_{1}X_{2}-1.630\;X_{1}X_{3}+0.817\;X_{2}X_{3} \end{array}$$
and the equation for 4,3-α-GTase (Equation (3)):

4,3-α-GTase Yield:(3)$$\begin{array}{c} =117.0+8.21\;X_{1}-0.075\;X_{2}-48.8\;X_{3}-0.001\;X_{1}^{2}-0.00172\;X_{2}^{2}+5.37\;X_{3}^{2}-\\ 0.0098\;X_{1}X_{2}-1.467\;X_{1}X_{3}+0.0358\;X_{2}X_{3} \end{array}$$

The results of the significance test of the regression coefficient and analysis of variance 4,6-α-GTase (Table 3) and 4,3-α-GTase (Table 4) showed the *p* = 0.0002 and *p* < 0.05; the *p* = 0.0152 and *p* < 0.05, respectively, for the regression model, and *p* = 0.1237 and *p* > 0.05; and the *p* = 0.7679 and *p* > 0.05, respectively, for the missing fitting term.

ANOVA results for the quadratic model of both GtfB enzymes showed that the *p*-values of the model were significant, but the *p*-values of the mismatch term were not. The results showed that the models of the two enzymes were established successfully and that the fitting degree of the selected models was statistically significant.

#### 3.2.2. Response Surface Analysis of Interaction of Various Factors on GtfB Production

The interaction between the two variables when the third variable is maintained at its optimum value is presented in the 2D contour plots and 3D surface plots in Figure 2.

The graphs indicate that the optimal yield of 4,6-α-GTase was highest at pH 6, which was supported by the initial substrate ratio of 1%. With regard to the air flow rate, the highest enzyme activity was observed at 10% and 50%, whereas it decreased at 30%. For 4,3-α-GTase, however, the air-flow rate had the opposite effect, with a higher enzyme yield observed at 30%. Regarding the initial substrate ratio, although the medium containing 5% glucose resulted in increased enzyme activity, this difference was not statistically significant (*p* > 0.05).

### 3.3. SEM Images of the Effect of 4,6-α-GTase and 4,3-α-GTase on Starch

To examine the effect of 4,6-α-GTase and 4,3-α-GTase enzymes on normal starch granules, image amplification ranging from 5000× to 60,000× was performed using Scanning Electron Microscopy (SEM). SEM images of the reaction medium (25 mM sodium acetate, 1 mM CaCl2, pH 5.5) containing 0.25% wheat, corn, and potato starch without enzyme are shown in Figure 3a,d,g.

SEM images of 4,3-α-GTase and 4,6-α-GTase enzymes after the (α1→4) bond hydrolase/transglycosylase activity was maintained in the reaction medium (25 mM sodium acetate, 1 mM CaCl_2_, pH 5.5) containing 0.25% wheat, corn, and potato starch concentrations are shown in Figure 3b,e,h and Figure 3c,f,i, respectively. The granule morphologies of enzyme-free wheat [34], corn [35], and potato [36] starches are smooth and unfragmented. The diameter of the unfragmented starch granules ranged from 871 nm to 5447 µm. The size of the starch granules in the enzyme-free wheat starch reaction was approximately 4–5 µm, which reduced to 100 nm after the 4,3-α-GTase reaction and to about 2 µm after the 4,6-α-GTase reaction. Similarly, the starch granules in the enzyme-free corn starch reaction were about 2 µm in size, with the number of granules decreasing to about 50 nm after the 4,3-α-GTase reaction and to about 2 µm after the 4,6-α-GTase reaction. Lastly, the starch granules in the enzyme-free potato starch reaction were roughly 2–3 µm in size, which reduced to 70–130 nm after the 4,3-α-GTase reaction and to about 500 nm after the 4,6-α-GTase reaction. Figure 3b,c show that starch granules exhibit a reduction in size upon interaction with 4,3-α-GTase and 4,6-α-GTase. This indicated that the enzyme acts on wheat starch and changes its morphology. As shown in Figure 3a,d,g, starch granules were quite uniform, whereas 4,3-α-GTase and 4,6-α-GTase enzymes acted on starch granules and showed more branching structures. This result is supported by the NMR results of our previous study, which revealed that it formed new (α1→3) and (α1→6) bonds [20].

### 3.4. Calorimetric DSC Properties of Bread and Bun Samples

In accordance with the experimental design, enthalpy values were determined using differential scanning calorimetry (DSC) analysis to evaluate the retarding properties of the enzymes produced within the scope of the study on staling for 7 days at 4 °C. The amylose content is known to be the main factor affecting the thermal properties of starch. The endothermic enthalpy values for the bread (Figure 4a) prepared with 4,6-α-GTase and 4,3-α-GTase, as well as the bun (Figure 4b), are displayed.

The enthalpy values increased during the storage period for all samples. It was observed on the third day that the enthalpy values of the breads produced using 4,6-α-GTase and 4,3-α-GTase were lower than those produced on production day (*p* < 0.05). On the third day, the endothermic enthalpy values of bread without enzyme, bread prepared with 4,6-α-GTase and bread prepared with 4,3-α-GTase were 294.9 J/g, 183.1 J/g and 180.2 J/g, respectively. According to these values, 4,6-α-GTase enzyme addition and 4,3-α-GTase enzyme addition significantly decreased the enthalpy value of normal bread by 37.9% and 38.9% on the third day, respectively (*p* < 0.05). On the seventh day, it was observed that the enthalpy values of the bread produced with the enzyme were lower than expected. On the seventh day, the endothermic enthalpy values of bread without enzyme, bread prepared with 4,6-α-GTase and bread prepared with 4,3-α-GTase were 326.3 J/g, 294.1 J/g and 254.3 J/g, respectively. A significant reduction in enthalpy was observed in breads produced with 4,3-α-GTase compared to the control bread (*p* < 0.05). These findings indicated that both 4,6-α-GTase and 4,3-α-GTase were effective in retarding the staling process of bread. The results indicated that 4,6-α-GTase was found to be more effective in preventing staling of bread than 4,3-α-GTase. Examination of the bun samples revealed that the enthalpy values were nearly identical. On the third day, the enthalpy of all the samples increased; however, no discernible distinction was observed among the bun samples, as the *p*-value surpassed the threshold of 0.05. On the seventh day, there was no discernible variation between the bread samples supplemented with 4,6-α-GTase and control bread. Conversely, bread supplemented with 4,3-α-GTase displayed dissimilarities compared to the others (*p* < 0.05). These results show that it was partially effective in delaying staling in the bun samples. It was found that the enthalpy of GtfB-modified starch gels decreased as the enzyme concentration increased from 1 to 4 U/g starch during the hydrolysis of GtfB, decreasing from 2.80 mJ/mg to 1.75 mJ/mg (*p* < 0.05) [17]. Our results showed that GtfB effectively decreased the retrogradation of bread prepared with 4,6-α-GTase and 4,3-α-GTase stored at 4 °C for 7 days.

### 3.5. Texture Characteristics of Bread and Bun Samples

The textural properties of the bread and bun prepared with 4,6-α-GTase and 4,3-α-GTase are listed in Table 5.

Hardness, stickiness, elasticity, and chewability were examined for their textural properties. Hardness increased significantly in all bread samples during storage (*p* < 0.05). The hardness of the bread made with 4,6-α-GTase and 4,3-α-GTase samples was lower on day seven, compared of the normal bread samples. The stickiness of all the bread samples decreased during storage. On the seventh day, the bread produced with 4,6-α-GTase and 4,3-α-GTase was stickier than the standard bread sample, and this result was statistically significant (*p* < 0.05). The elasticity of bread containing enzymes was significantly greater (*p* < 0.05). Furthermore, the breads exhibited reduced chewability, requiring less effort to swallow, as indicated by the significant decrease in chewing energy (*p* < 0.05). The results presented herein indicate that enzymes alter the structural characteristics of bread by influencing the configuration of starch. The starch amylose fraction exhibited significant changes, including a reduction in size, an increase in stickiness, a decrease in hardness, and a decrease in chewing energy.

The observation was made that the hardness of the bun products increased during storage. On the seventh day, the hardness of the samples that had been treated with enzymes was found to be significantly lower (*p* < 0.05). It was observed that stickiness decreased in all bun products. However, it was determined that the stickiness was higher in 4,3-α-GTase supplemented breads (*p* < 0.05). In terms of elasticity, it was discovered that the addition of 4,3-α-GTase resulted in higher elasticity values for bun. The samples displayed no discernible disparities in chewability. These findings suggest that the influence of 4,3-α-GTase and 4,6-α-GTase on the bun structure is restricted. It is noteworthy that differences in the characteristics of stickiness and chewiness in bread were not observed in the bun. This may be because of the addition of eggs, sugar, and fat, as required by the formulation for bun production.

### 3.6. eGI and SDS Properties of Bread and Bun Samples

Starch is the second most abundant carbohydrate on Earth and is a significant dietary source of carbohydrates for humans [37]. The incidence of diet-related diseases, such as obesity and diabetes, has increased globally, leading to a focus on dietary approaches [38]. For the purpose of nutrition, starch has been categorized into three types based on its digestion properties: rapidly digestible starch (RDS), slowly digestible starch (SDS), and resistant starch (RS) [25]. The RDS, SDS, and RS contents calculated using the Englyst procedure are listed in Table 6.

The SDS content was found to increase significantly in breads prepared with 4,3-α-GTase and 4,6-α-GTase (*p* < 0.05). Amylopectin is the primary constituent of SDS, and the optimal molecular structure of amylopectin for SDS can be achieved through chemical, physical, enzymatic, and genetic modifications. In general, (α1→6) glycosidic bonds are more resistant to cleavage than (α1→4) bonds. Therefore, the presence of a higher number of (α1→6) bonds in starch would impede its digestion [39].

Similarly, the RS content of the breads prepared with 4,6-α-GTase and 4,3-α-GTase increased, while the RDS content decreased. In the comparison between both enzymes, 4,6-α-GTase resulted in higher SDS and RS content in bread than 4,3-α-GTase, whereas 4,3-α-GTase resulted in higher RDS content than 4,6-α-GTase. SDS, a starch fraction between RDS and RS, is slowly digested in the small intestine for sustained glucose release with low initial glycemia [40]. Both enzymes (4,6-α-GTase and 4,3-α-GTase) significantly decreased the glycemic index of breads. Accordingly, the estimated glycemic index (eGI) of bread prepared with 4,6-α-GTase was reduced by 18.01% while eGI of bread prepared with 4,3-α-GTase was reduced by 13.61%. The eGI values were higher for bun products than for breads. It is likely that other additives in bun production also have an effect. In particular, the addition of sugar might have caused this increase. Bakery products, such as bread and buns, are considered to have a high glycemic index (>70). Accordingly, bread and bun products produced with 4,6-α-GTase and 4,3-α-GTase enzyme additives remained at a value of 70 and above, which is considered high glycemic index, but this value could still be reduced. Although 4,6-α-GTase and 4,3-α-GTase caused a significant decrease in the glycemic index of both bread and buns, it is understood that higher enzyme concentrations should be used in order not to be considered as high glycemic index products [39].

Similar results were obtained for bun samples. However, a lower RS, higher RDS, and lower SDS were observed in the bun samples. In addition, the rate of reduction of eGI with enzyme treatment was similar to that of bread in the bun samples. The eGI could be reduced by the 4,6-α-GTase and 4,3-α-GTase enzymes by forming glycosidic bond branching in the starch amylose fraction. The enzyme 4,6-α-GTase cleaves (α1→4) linkages and forms new consecutive (α1→6) linkages, whereas 4,3-α-GTase cleaves (α1→4) linkages and forms new consecutive (α1→3) linkages [7,14]. It was concluded that the enzymes 4,6-α-GTase and 4,3-α-GTase resulted in the formation of more branched structures, which led to an increase in the amount of amylopectin and, subsequently, SDS content. The process of starch retrogradation, which involves the reorganization of amylose and amylopectin molecules into double helices and potential crystalline structures, is a crucial phenomenon that occurs during food storage, particularly at cold temperatures [39]. Enhancing the quantity of SDS, and subsequently the branching of amylopectin, resulted in a further reduction in bread retrogradation and a delay in bread staling. The RDS content of StGtfB-modified starch decreased significantly to 43.69% (*p* < 0.05), whereas the SDS content increased significantly to 53.22% (*p* < 0.05), as the enzyme concentration increased to 5 U/g starch during the hydrolysis of StGtfB [41]. In our study, the SDS content of bread prepared with 4,6-α-GTase and 4,3-α-GTase increased significantly by 49.5% and 38% (*p* < 0.05), respectively, while the RDS content decreased to 61.27% and 49.58%, respectively. In another study, hydrolysis using GtfB at 1–4 U/g starch, the amylose content of the GtfB-modified starches decreased significantly to 16.34–17.11% (*p* < 0.05), whereas the amylopectin content increased to 76.55–77.65% (*p* < 0.05) [17].

### 3.7. Sensory Properties of Bread and Bun Samples

The data used to illustrate the RATA feature relate to a rate-all-that-apply (RATA) experiment, in which 10 assessors evaluated 10 attributes over one session with a total of six samples, comprising three breads and three buns prepared with 4,6-α-GTase and 4,3-α-GTase, along with a control sample (Table 7).

In this table, it can be seen that, for all attributes except Pore Structure, Internal Texture, Swallowability, Fermented Odor and General Evaluation, the product variable is significant at the 5% threshold, indicating that these attributes are discriminating.

Another important result of the RATA feature is the graphical representation of products and attributes, as shown in Figure 5. In the following chart, it can be seen that we have 95.24% inertia reported by our first two axes; therefore, we can be satisfied with these two. The chart shows an opposition between wetness–dryness, softness and other attributes. Breads prepared with 4,6-α-GTase and 4,3-α-GTase were more discriminative in terms of color than the control bread, while the control bread was more discriminative in terms of acid odor than the breads prepared with the enzymes. The pore structure, internal texture, swallowability, fermented odor, and general evaluation, which are statistically non-discriminative characteristics, are presented at a certain point in Figure 5. In terms of buns, chewiness was positively correlated with the control bun, whereas it was negatively correlated with softness and wetness–dryness. However, buns prepared with 4,6-α-GTase and 4,6-α-GTase showed a positive correlation with softness and wetness–dryness.

These results showed that the use of 4,6-α-GTase and 4,3-α-GTase did not affect the sensory properties of bakery products, such as bread and bun. These enzymes, which have the ability to modify starch, did not affect the structural and flavor properties of bread.

## 4. Conclusions

In our previous research, we successfully accomplished the heterologous extracellular production of 4,6-α-GTase and 4,3-α-GTase, which belong to the GH70 enzyme family, in *Lactococcus lactis*. In *L. lactis*, the continuous promoter p45 and part of the usp45 signaling series containing the catalytic site of the enzymes were shown to be successful in secretion from the cell. The recombinant *L. lactis* strains obtained in that study are candidate microorganisms for the industrial production of 4,6-α-GTase and 4,3-α-GTase. In particular, the GRAS qualities of these strains offer great advantages for their production in the food industry. Therefore, these strains are recommended for industrial application.

Here in RSM optimization studies, 4,6-α-GTase and 4,3-α-GTase enzymes were able to be produced in the highest amount (15.63 ± 1.65 and 19.01 ± 1.75 U/mL) at 1% glucose, pH 6, and 30 °C, respectively. In line with the study objectives, the optimal conditions for the enzyme and its production in the batch system were optimized. However, to achieve more efficient enzyme production on an industrial scale, increasing the production capacity of these recombinant strains is recommended for further studies. In particular, attempts have been made to use different substrate sources for the production of these enzymes.

It has been demonstrated that 4,6-α-GTase and 4,3-α-GTase produced by *L. lactis* by extracellular secretion have technological and health functional effects in bakery products, such as bread and bun. These enzymes successfully lowered the glycemic index and delayed staling, which were the objectives of the present study. Innovative enzymes have been developed for use in food technologies. It was observed that the estimated glycemic index (eGI) of bread prepared with 4,6-α-GTase was reduced by 18.01% while eGI of bread prepared with 4,3-α-GTase was reduced by 13.61%, and staling could be delayed.

A well-functioning recombinant host was developed for the industrial production of enzymes that modify the starch structure of bakery products, and the effects of these enzymes were verified and successfully demonstrated for use in bakery products. These enzymes will enable the production of healthy bakery products in food technology, as well as the efficient use of resources, by increasing product quality and shelf life.

## Figures and Tables

**Figure 1 foods-13-00432-f001:**
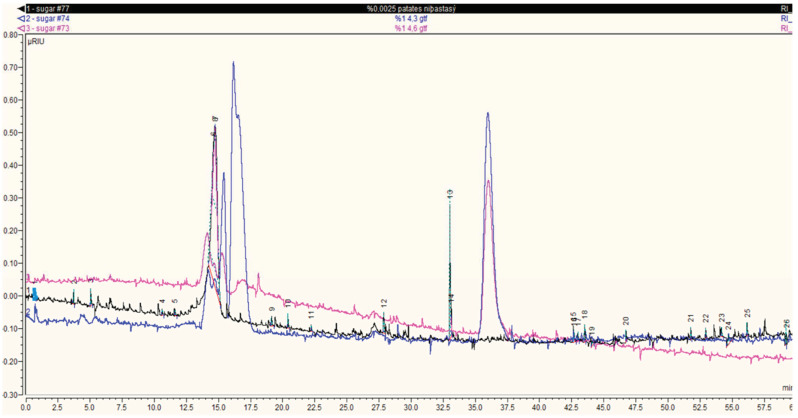
High-pressure size exclusion chromatography (HPSEC) results of the molecular size and structural characteristics of the products formed by the catalytic action of 4,6-α-GTase and 4,3-α-GTase enzymes on potato starch. Potato starch reaction without enzyme (black), 4,6-α-GTase and potato starch reaction (pink), 4,3-α-GTase enzyme and potato starch reaction (blue).

**Figure 2 foods-13-00432-f002:**
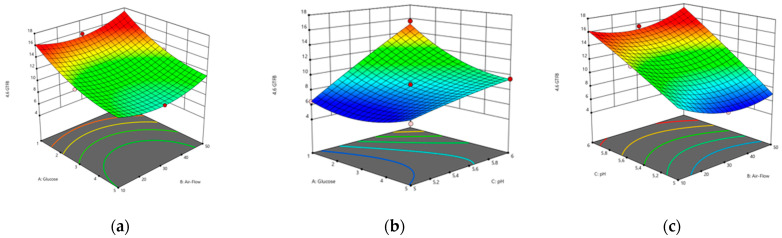
Response surface analysis diagram of the interaction of various factors on GtfB production. (**a**) Effects of glucose and air-flow on 4,6-α-GtfB production; (**b**) effects of glucose and pH on 4,6-α-GtfB production; (**c**) effects of pH and air-flow on 4,6-α-GtfB production; (**d**) effects of glucose and air-flow on 4,3-α-GtfB production; (**e**) effects of glucose and pH on 4,3-α-GtfB production; (**f**) effects of pH and air-flow on 4,3-α-GtfB production.

**Figure 3 foods-13-00432-f003:**
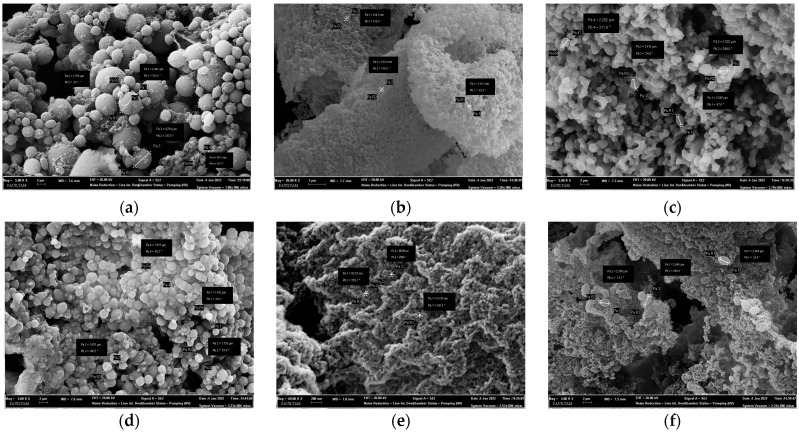
SEM images ((**a**) 5000×, (**b**) 20,000×, (**c**) 5000×-image indicating granule sizes) of enzyme-free (**a**), 4,3-α-Gtfb enzyme-containing (**b**), and 4,6-α-Gtfb enzyme-containing (**c**) reaction media (25 mM sodium acetate and 1 mM CaCl_2_, pH 5.5) containing 0.25% wheat starch. SEM images ((**d**) 5000×, (**e**) 60,000×, (**f**) 5000×-image indicating granule sizes) of enzyme-free (**d**), 4,3-α-Gtfb enzyme-containing (**e**), and 4,6-α-Gtfb enzyme-containing (**f**) reaction media (25 mM sodium acetate and 1 mM CaCl_2_, pH 5.5) containing 0.25% corn starch. SEM images ((**g**) 5000×, (**h**) 60,000×, (**i**) 20,000×-image indicating granule sizes) of enzyme-free (**g**), 4,3-α-Gtfb enzyme-containing (**h**) and 4,6-α-Gtfb enzyme-containing (**i**) reaction media (25 mM sodium acetate, 1 mM CaCl_2_ pH 5.5) containing 0.25% potato starch concentration.

**Figure 4 foods-13-00432-f004:**
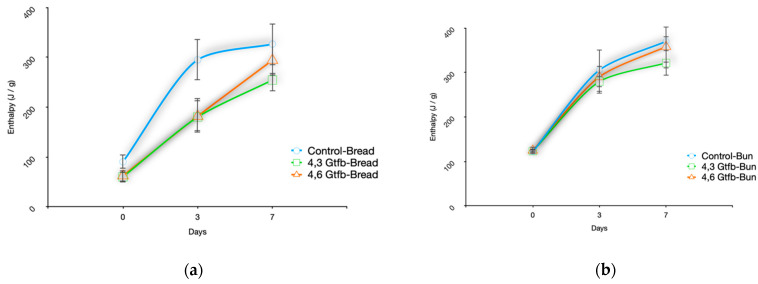
Enthalpy changes of bread and bun samples prepared with 4,6-α-GTase (**a**) and 4,3-α-GTase (**b**) during storage.

**Figure 5 foods-13-00432-f005:**
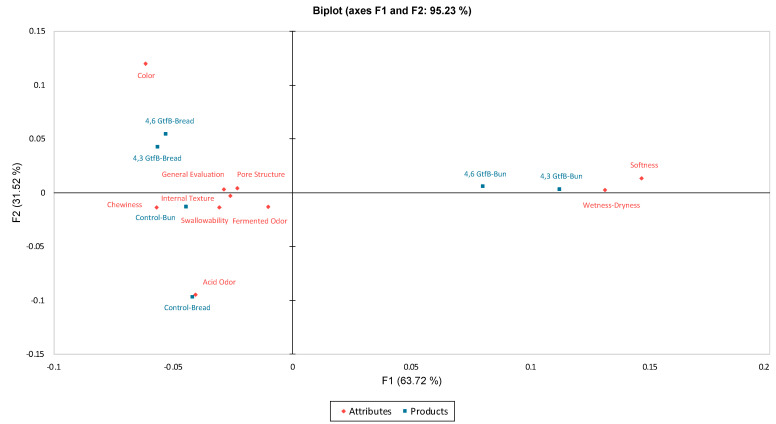
The PCA Biplot result of the RATA feature is a graphical representation of the products and attributes regarding sensory analysis.

**Table 1 foods-13-00432-t001:** Levels of each factor in the Box–Behnken experimental design.

		Variables and Ranges		
Factors	Independent Variables	−1	0	+1
** $X_{1}$ **	Initial substrate concentration (g dm^−3^) %	1%	3%	5%
** $X_{2}$ **	Air-flow rate (vvm) %	10	30	50
** $X_{3}$ **	pH	5	5.5	6

**Table 2 foods-13-00432-t002:** Results of the Box–Behnken design.

Run	Factors			4,6-α-GTFB		4,3-α-GTFB	
	$X_{1}$ (Glu)	$X_{2}$ (O_2_)	$X_{3}$ (pH)	Predicted 4,6-α-GTase Production (U/mL)	4,6-α-GTase Production (U/mL)	Predicted 4,3-α-GTase Production (U/mL)	4,3-α-GTase Production (U/mL)
**1**	−1	0	+1	15.18	15.63 ± 1.65	18.8	19.01 ± 1.75
**2**	−1	0	−1	6.5	6.46 ± 0.43	9.03	9.83 ± 0.71
**3**	+1	−1	0	10.67	11.03 ± 1.61	11.85	12.47 ± 0.91
**4**	0	+1	−1	6.09	6.49 ± 0.54	8.92	8.74 ± 0.65
**5**	0	0	0	8.52	8.35 ± 0.11	12.26	10.86 ± 0.37
**6**	0	−1	−1	8.43	8.53 ± 0.73	10.06	9.65 ± 0.99
**7**	0	0	0	8.52	8.46 ± 0.82	12.26	12.03 ± 0.74
**8**	0	+1	+1	13.15	13.06 ± 1.25	16.47	16.89 ± 2.06
**9**	−1	+1	0	12.44	12.09 ± 1.21	12.06	11.45 ± 1.14
**10**	0	0	0	8.52	8.76 ± 0.93	12.26	13.89 ± 1.84
**11**	+1	0	−1	7.29	6.85 ± 0.35	11.32	11.12 ± 0.42
**12**	−1	−1	0	12.85	12.81 ± 1.66	11.70	11.32 ± 0.5
**13**	+1	+1	0	9.68	9.73 ± 0.91	10.64	11.03 ± 0.23
**14**	0	−1	+1	12.22	11.83 ± 0.65	16.18	16.37 ± 1.33
**15**	+1	0	+1	9.45	9.5 ± 0.98	15.23	14.43 ± 0.21

**Table 3 foods-13-00432-t003:** ANOVA Results for Quadratic model of 4,6-α-GTase.

Source of Variance	Sum of Squares	df	Mean Square	F-Value	*p*-Value	
**Model**	100.54	9	11.17	52.28	0.0002	significant
** $X_{1}$ **	12.20	1	12.20	57.10	0.0006	
** $X_{2}$ **	1.00	1	1.00	4.69	0.0827	
** $X_{3}$ **	58.81	1	58.81	275.21	<0.0001	
** $X_{1}X_{2}$ **	0.0841	1	0.0841	0.3936	0.5580	
** $X_{1}X_{3}$ **	10.63	1	10.63	49.74	0.0009	
** $X_{2}X_{3}$ **	2.67	1	2.67	12.51	0.0166	
** $X_{1}^{2}$ **	5.88	1	5.88	27.52	0.0033	
** $X_{2}^{2}$ **	9.81	1	9.81	45.89	0.0011	
** $X_{3}^{2}$ **	0.1136	1	0.1136	0.5317	0.4986	
**Residual error**	1.07	5	0.2137			
**Lack of fit**	0.9783	3	0.3261	7.24	0.1237	not significant
**Pure error**	0.0901	2	0.0450			
**Total deviation**	101.61	14				

**Table 4 foods-13-00432-t004:** ANOVA Results for Quadratic model of 4,3-α-GTase.

Source of Variance	Sum of Squares	df	Mean Square	F-Value	*p*-Value	
**Model**	113.47	9	12.61	8.40	0.0152	significant
** $X_{1}$ **	0.8192	1	0.8192	0.5459	0.4932	
** $X_{2}$ **	0.3612	1	0.3612	0.2407	0.6445	
** $X_{3}$ **	93.57	1	93.57	62.35	0.0005	
** $X_{1}X_{2}$ **	0.6162	1	0.6162	0.4106	0.5499	
** $X_{1}X_{3}$ **	8.61	1	8.61	5.74	0.0619	
** $X_{2}X_{3}$ **	0.5112	1	0.5112	0.3406	0.5848	
** $X_{1}^{2}$ **	0.0001	1	0.0001	0.0000	0.9955	
** $X_{2}^{2}$ **	1.75	1	1.75	1.17	0.3293	
** $X_{3}^{2}$ **	6.64	1	6.64	4.43	0.0893	
**Residual error**	7.50	5	1.50			
**Lack of fit**	2.83	3	0.9446	0.4046	0.7679	not significant
**Pure error**	4.67	2	2.33			
**Total deviation**	120.98	14				

**Table 5 foods-13-00432-t005:** Textural properties of bread and bun samples prepared with and 4,6-α-GTase and 4,3-α-GTase.

Days	Control Bread	4,6 GtfB-Bread	4,3 GtfB-Bread	Control-Bun	4,6 GtfB-Bun	4,3 GtfB-Bun
**Hardness (N)**						
**0**	3.71 ± 0.30 ^c, A^	3.66 ± 0.38 ^c, A^	3.36 ± 0.32 ^c, A^	3.71 ± 0.30 ^c, A^	2.45 ± 0.62 ^c, B^	1.91 ± 0.48 ^c, B^
**3**	8.44 ± 0.22 ^b, A^	6.70 ± 0.70 ^b, B^	7.18 ± 0.58 ^b, A^	8.44 ± 0.22 ^b, A^	7.45 ± 0.75 ^b, A^	6.55 ± 1.84 ^b, B^
**7**	20.24 ± 5.39 ^a, A^	14.09 ± 1.98 ^a, B^	12.02 ± 1.75 ^a, B^	20.24 ± 5.39 ^a, A^	16.19 ± 5.11 ^a, B^	11.53 ± 2.84 ^a, B^
**Stickiness (c)**						
**0**	0.74 ± 0.04 ^a, A^	0.78 ± 0.02 ^a, A^	0.81 ± 0.01 ^a, A^	0.74 ± 0.03 ^a, B^	0.82 ± 0.01 ^a, A^	0.82 ± 0.03 ^a, A^
**3**	0.48 ± 0.03 ^b, A^	0.48 ± 0.02 ^b, A^	0.50 ± 0.06 ^b, A^	0.57 ± 0.04 ^b, B^	0.55 ± 0.02 ^b, B^	0.61 ± 0.06 ^b, A^
**7**	0.39 ± 0.04 ^c, B^	0.47 ± 0.04 ^b, A^	0.44 ± 0.03 ^b, A^	0.39 ± 0.05 ^c, B^	0.41 ± 0.04 ^c, A^	0.43 ± 0.06 ^c, A^
**Elasticity (mm)**						
**0**	8.22 ± 0.18 ^a, A^	7.92 ± 0.19 ^a, B^	8.16 ± 0.21 ^a, A^	8.22 ± 0.18 ^a, A^	8.30 ± 0.12 ^a, A^	8.32 ± 0.15 ^a, A^
**3**	7.63 ± 0.30 ^b, B^	7.88 ± 0.20 ^a, B^	8.23 ± 0.11 ^a, A^	7.63 ± 0.30 ^b, B^	8.17 ± 0.16 ^a, A^	8.04 ± 0.18 ^a, A^
**7**	7.25 ± 0.45 ^b, B^	7.80 ± 0.29 ^a, A^	8.06 ± 0.17 ^a, A^	7.24 ± 0.45 ^b, B^	7.65 ± 0.19 ^b, B^	8.01 ± 0.15 ^a, A^
**Chewability (Mj)**						
**0**	22.58 ± 1.41 ^c, A^	22.08 ± 1.33 ^c, A^	15.59 ± 2.25 ^c, B^	22.58 ± 1.40 ^c, A^	14.59 ± 2.28 ^b, B^	13.14 ± 3.79 ^c, B^
**3**	36.64 ± 2.57 ^b, A^	30.73 ± 2.30 ^b, B^	34.20 ± 3.20 ^b, A^	36.64 ± 2.57 ^b, A^	33.58 ± 1.66 ^a, A^	31.49 ± 6.57 ^b, A^
**7**	55.07 ± 9.35 ^a, A^	51.45 ± 5.77 ^a, A^	40.80 ± 8.01 ^a, B^	55.07 ± 9.34 ^a, A^	36.44 ± 10.93 ^a,B^	55.95 ± 19.97 ^a, A^

Different lowercase letters indicate the difference between storage days (*p* < 0.05), and uppercase letters indicate the difference between sample groups (*p* < 0.05).

**Table 6 foods-13-00432-t006:** Starch content and estimated glycemic index characteristics of bread and bun samples prepared with 4,6-α-GTase and 4,3-α-GTase.

	RS (g/100 g)	RDS (g/100 g)	SDS (g/100 g)	HI	eGI
**Control Bread**	3.45 ± 0.75 ^c^	32.25 ± 0.81 ^a^	22.95 ± 1.10 ^b^	93.08 ± 5.50 ^a^	90.81 ± 7.99 ^a^
**4,6 GtfB-Bread**	9.87 ± 0.52 ^b^	12.49 ± 0.58 ^b^	34.31 ± 0.06 ^a^	63.27 ± 7.48 ^b^	74.45 ± 4.18 ^b^
**4,3 GtfB-Bread**	11.41 ± 0,75 ^a^	16.56 ± 0.62 ^c^	31.67 ± 0.94 ^a^	70.08 ± 6.84 ^b^	78.18 ± 5.09 ^b^
**Control Bun**	3.76 ± 0.75 ^b^	26.14 ± 2.82 ^a^	15.88 ± 1.37 ^b^	127.85 ± 10.64 ^a^	109.89 ± 12.25 ^a^
**4,6 GtfB Bun**	5.17 ± 0.86 ^a^	19.97 ± 2.44 ^b^	17.81 ± 1.18 ^a^	91.45 ± 11.99 ^b^	89.67 ± 2.76 ^b^
**4,3 GtfB Bun**	6.12 ± 1.22 ^a^	14.53 ± 1.90 ^c^	18.94 ± 1.54 ^a^	87.49 ± 12.06 ^b^	87.74 ± 3.79 ^b^

Different lowercase letters indicate the difference between storage days (*p* < 0.05).

**Table 7 foods-13-00432-t007:** The data relate to a rate-all-that-apply (RATA) experiment, in which 10 assessors evaluated 10 attributes over one session, with a total of six samples, comprising three breads and three buns prepared with 4,6-α-GTase and 4,3-α-GTase.

		Color	Pore Structure	Internal Texture	Chewiness	Softness	Wetness-Dryness	Swallowability	Fermented Odor	Acid Odor	General Evaluation
**R^2^**		0.5916	0.7570	0.5087	0.5495	0.6320	0.7067	0.6243	0.8611	0.4984	0.4295
**F**		4.6571	10.0166	3.3286	3.9214	5.5215	7.7483	5.3423	19.9285	3.1943	2.4206
**Pr > F**		<0.0001	<0.0001	0.001	0.000	<0.0001	<0.0001	<0.0001	<0.0001	0.002	0.013
**Product**	F	4.7956	1.4651	0.6047	2.7	8.1534	8.8807	0.3103	0	3.8571	0.4444
	Pr > F	0.001	0.220	0.697	0.032	<0.0001	<0.0001	0.904	1.000	0.005	0.815
**Assessor**	F	4.5802	14.7674	4.8418	4.6	4.05940	7.1192	8.1379	31	2.8260	3.5185
	Pr > F	0.000	<0.0001	0.000	0.000	0.001	<0.0001	<0.0001	<0.0001	0.010	0.002

## Data Availability

Data are contained within the article.

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
