# Peer review of "Optimization of 4,6-α and 4,3-α-Glucanotransferase Production in Lactococcus lactis and Determination of Their Effects on Some Quality Characteristics of Bakery Products"

_foods, 2024, doi:10.3390/foods13030432_

Round 1

Reviewer 1 Report

Comments and Suggestions for Authors

It is significant to use the glucanotransferase isolated fron the two strains of Lactococus for barkery products, and the data obtained are important in the technical improvements in the future.

There are needs some improve and revison to the MS.

Comments on the Quality of English Language

The English qulity of the MS needs small revisions

Author Response

Response to Reviewer 1 Comments

1. Summary

2. Questions for General Evaluation

Reviewer’s Evaluation

Response and Revisions

Does the introduction provide sufficient background and include all relevant references?

Can be improved

Thank you for your comment. We improved the introduction and all relevant references.

Are all the cited references relevant to the research?

Yes

Is the research design appropriate?

Yes

Are the methods adequately described?

Yes

Are the results clearly presented?

-

Are the conclusions supported by the results?

Can be improved

Thank you for your comment. We improved the conclusions.

3. Point-by-point response to Comments and Suggestions for Authors

Comments 1: There are needs some improve and revison to the MS.

Response 1: Thank you for pointing this out. We agree with this comment. As you clearify in the text that you have upload the system, we have made all corrections you highlighted and we upload the revised file. Thank you for your effort and kindly evaluation.

4. Response to Comments on the Quality of English Language

Point 1: There are needs some improve and revison to the MS.

Response 1: Native english speaker helped us to improve language of manuscrirpt and other tools were used for grammar check.

Reviewer 2 Report

Comments and Suggestions for Authors

The research paper focuses on optimizing the production of 4, 6-α and 4, 3-α- GTase in Lactococcus lactis and examining their impact on bakery products. The study is innovative in its approach to reducing the glycemic index and delaying staling in bakery items, employing enzymes derived from L. lactis. The experimental design appears well-structured, utilizing methods like High-Pressure Size Exclusion Chromatography (HP-SEC) and Response Surface Methodology (RSM) for optimization. The application prospects of this research are significant, especially in the health-conscious food industry, aiming to produce bakery products with lower glycemic indices and enhanced shelf life. But there are still some questions that need to be answered.

Question:

1. Line 162: I didn’t see the SEM image with a magnification of 3000 in the results.

2. Line 72: Please ensure the correctness of the SI unit symbols.

3. Line 241: Please make sure the fonts are consistent.

4. Line 429-457: Due to the non-uniform scale of the images, it was not possible to confirm the reduction in the size of starch granules described by the authors in the different treatment groups. Please give a clear SEM image and a specific explanation of the changes in starch granule size for different experimental groups.

5. Figure 3: The image quality is too low. Please provide a clearer image, and ensure the image includes a clear magnification factor and scale bar.

6. Line 494: Please check for missing punctuation marks.

7. Line 563-565: What evidence is there to support the point you make here? Please give specific basis.

8. Please check the table in the full text and make sure the format is consistent.

Comments on the Quality of English Language

The English language should be improved by a native English speaker. 

Author Response

Response to Reviewer 2 Comments

1. Summary

2. Point-by-point response to Comments and Suggestions for Authors

Comments 1: Line 162: I didn’t see the SEM image with a magnification of 3000 in the results.

Response 1: Thank you for pointing this out. We agree with this comment. It has been deleted magnification of 3000x.

Comments 2: Line 72: Please ensure the correctness of the SI unit symbols.

Response 2: Thank you. We have, accordingly, done of the SI Unit symbols.

Comments 3: Line 241: Please make sure the fonts are consistent.

Response 3: Thank you. We have corrected the fonts.

Comments 4: Line 429-457: Due to the non-uniform scale of the images, it was not possible to confirm the reduction in the size of starch granules described by the authors in the different treatment groups. Please give a clear SEM image and a specific explanation of the changes in starch granule size for different experimental groups.

Response 4: Thank you for your comment. We revised and added a specific explantion of the changes in starch granule size for different experimental groups.

“The size of the starch granules in the enzyme-free wheat starch reaction was approxi-mately 4-5 µm, which reduced to 100 nm after the 4,3--GTase reaction and to about 2 µm after the 4,6--GTase reaction. Similarly, the starch granules in the enzyme-free corn starch reaction were about 2 µm in size, with the number of granules decreasing to about 50 nm after the 4,3--GTase reaction and to about 2 µm after the 4,6--GTase reaction. Lastly, the starch granules in the enzyme-free potato starch reaction were roughly 2-3 µm in size, which reduced to 70-130 nm after the 4,3--GTase reaction and to about 500 nm after the 4,6--GTase reaction. “

Comments 5: Figure 3: The image quality is too low. Please provide a clearer image, and ensure the image includes a clear magnification factor and scale bar.

Response 5: Thank you. We changed current images with high resolution .tif images. Please check.

Comments 6: Line 494: Please check for missing punctuation marks.

Response 6: Thank you. We checked for missing punctuation markes and we corrected.

Comments 7: Line 563-565: What evidence is there to support the point you make here? Please give specific basis.

Response 7: Thank you for your comment. We think we could not able to express these findings clearly. We revised these sentences as “Accordingly, bread and bun products produced with 4,6--GTase and 4,3--GTase en-zyme additives remained at a value of 70 and above, which is considered high glycemic index, but this value could still be reduced. Although 4,6--GTase and 4,3--GTase caused a significant decrease in the glycemic index of both bread and buns, it is under-stood that higher enzyme concentrations should be used in order not to be considered as high glycemic index products [39].” and we added the literature.

Comments 8: Please check the table in the full text and make sure the format is consistent.

Response 8: Thank you. We checked and corrected the tables in the full text.

4. Response to Comments on the Quality of English Language

Point 1: The English language should be improved by a native English speaker.

Response 1: Native english speaker helped us to improve language of manuscrirpt and other tools were used for grammar check.

Reviewer 3 Report

Comments and Suggestions for Authors

In current paper the production of 4,6 (4,6-GTase) and 4,3-glucanotransferase (4,3- 13

GTase) expressed previously in Lactococcus lactis was optimized and these enzymes were used to

investigate glycemic index reduction and staling delay in bakery products. HP-SEC analysis showed

that the relevant enzymes were able to produce oligosaccharides from potato starch or maltooligo- saccharides.

Comments good tests, good results and good graphics plus 2 D diagrams AND PICTURES and Response Surface Methodology (RSM) to optimize enzyme synthesis and the  highest enzyme activities please check accuracy in English language

Comments on the Quality of English Language

MINOR

Author Response

Response to Reviewer 3 Comments

1. Summary

2. Questions for General Evaluation

Reviewer’s Evaluation

Response and Revisions

Does the introduction provide sufficient background and include all relevant references?

Can be improved

Thank you for your comment. We improved the introduction and all relevant references.

Are all the cited references relevant to the research?

Can be improved

Thank you for your comment. We checked all cited references relevant to the research.

Is the research design appropriate?

Yes

Are the methods adequately described?

Yes

Are the results clearly presented?

Yes

Are the conclusions supported by the results?

Yes

3. Response to Comments on the Quality of English Language

Point 1: please check accuracy in English language

Response 1: Native english speaker helped us to improve language of manuscrirpt and other tools were used for grammar check.
